# Impact of a Novel Pretreatment on Bond Strength of Universal Adhesive to Conventional and CAD/CAM Resin Composites: In Vitro Study

**DOI:** 10.3390/jfb16060197

**Published:** 2025-05-27

**Authors:** Ali A. Elkaffas, Abdullah Alshehri, Feras Alhalabi, Rania Bayoumi, Abdullah Ali Alqahtani, Abdulellah Almudahi, Abdulaziz Fahd Alsubaie, Abdulaziz Fahd Alharbi

**Affiliations:** 1Conservative Dental Science Department, College of Dentistry, Prince Sattam bin Abdulaziz University, Al-Kharj 11942, Saudi Arabia; a.elkaffas@psau.edu.sa (A.A.E.); am.alshehri@psau.edu.sa (A.A.); r.gad@psau.edu.sa (R.B.); aa.alqahtani@psau.edu.sa (A.A.A.); a.almudahi@psau.edu.sa (A.A.); 2Department of Operative Dentistry, Faculty of Dentistry, Mansoura University, Mansoura 35516, Egypt; 3Biomaterials Department, Faculty of Dental Medicine for Girls, Al-Azhar University, Cairo 11754, Egypt; 4College of Dentistry, Prince Sattam bin Abdulaziz University, Al-Kharj 11942, Saudi Arabia; 439050409@std.psau.edu.sa (A.F.A.); im__3zooz@hotmail.com (A.F.A.)

**Keywords:** copper sulfate, dipotassium hydrogen phosphate, CAD/CAM resin blocks, universal adhesives

## Abstract

Novel dentin bonding pretreatment using copper sulfate (CuSO_4_) and dipotassium hydrogen phosphate (K_2_HPO_4_) may create a more hydrophobic environment for dentin bonding. Thus, this study aims to investigate the impact of a CuSO_4_ + K_2_HPO_4_ pretreatment on dentin μTBS when bonded with a universal adhesive to conventional and CAD/CAM resin composites. Eighty recently extracted human molars (n = 80) were chosen and placed in transparent acrylic blocks to expose the crowns entirely. Nano-filled resin composite and CAD/CAM resin blocks were selected. Based on the dentin pretreatment, type of resin composite, and adhesion strategy, the teeth were randomly allocated into eight equal groups (n = 10). The microtensile bond strength (μTBS) and fracture mode were determined. A three-way analysis of variance (ANOVA) was used to analyze the μTBS data, followed by Tukey’s post hoc test. The μTBS values were not significantly affected by either the resin composite type (*p* > 0.05) or the adhesive strategy (*p* > 0.05) according to the three-way ANOVA results. Conversely, significant differences were detected between no dentin pretreatment (24.20 ± 4.54 MPa) and CuSO_4_ + K_2_HPO_4_ pretreatment (33.66 ± 5.22 MPa) using an etch-and-rinse adhesive strategy for nano-filled composites (*p* < 0.001). Additionally, significant differences were detected between no dentin pretreatment (24.71 ± 4.33 MPa) and CuSO_4_ + K_2_HPO_4_ pretreatment (32.49 ± 4.92 MPa) using an etch-and-rinse adhesive strategy for CAD/CAM resin blocks (*p* < 0.001). Moreover, significant differences were detected between no dentin pretreatment (21.20 ± 3.40 MPa) and CuSO_4_ + K_2_HPO_4_ pretreatment (30.31 ± 3.87 MPa) using a self-etching adhesive strategy for nano-filled composites (*p* < 0.001). Also, significant differences were detected between no dentin pretreatment (23.89 ± 3.89 MPa) and CuSO_4_ + K_2_HPO_4_ pretreatment (31.22 ± 4.71 MPa) using a self-etching adhesive strategy for CAD/CAM resin blocks (*p* < 0.001). In conclusion, dentin μTBS was enhanced by a copper-based treatment when used with nano-filled and CAD/CAM resin blocks.

## 1. Introduction

The discipline of adhesive dentistry has significantly advanced over recent decades. This innovative approach advocates a more conservative cavity design reliant on the efficacy of contemporary simplified adhesives within the confines of minimally invasive dentistry. Three pretreatment stages have been amalgamated into a single bottle to simplify adhesive systems. These so-called “universal adhesives” include solvents, initiators, functional and base monomers, and optional fillers. The term “universal” signifies the adhesives’ compatibility with etch-and-rinse, self-etching, and selective enamel etching application modalities. In vitro and clinical studies indicate that selective enamel etching produces optimal results without affecting dentin when using universal adhesives [1,2]. An innovative universal adhesive, Scotchbond Universal Plus, has been developed; it is the first universal adhesive exhibiting radiopacity comparable to dentin [3].

Dentin bonding is characterized by insufficient resin monomer infiltration into the demineralized dentin matrix and the ensuing degradation of collagen and resin monomers, adversely affecting the quality of the hybrid layer [4]. Wet bonding utilizing ethanol [5], using an electric current to enhance resin infiltration [6], integrating crosslinking agents [7], eliminating proteoglycans [8], and other approaches have all been investigated as potential ways to increase the permeability of adhesive monomers to demineralized dentin [1]. Nevertheless, additional study is necessary because no technique has yet been able to fully address the underlying issues of dentin bonding.

Metals added to dentin bonding materials may make them more hydrophobic. Copper, a bivalent metal, reduces collagen breakdown by inhibiting matrix metalloproteinase (MMP). By acting as an indirect cross-linking agent, copper strengthens the collagen network [9]. Moreover, both gram-negative and gram-positive bacteria are effectively inhibited by copper’s antibacterial capabilities [4]. These benefits reflect copper’s significant application value and promising development potential in dentin bonding [9].

Dental prosthodontics, orthodontic equipment, dental implants, and restorative materials are all potential applications for nanoparticles containing copper in the dental field. The toxicity of nanoparticles containing copper has prompted researchers to look for less harmful versions of the material [10]. Actually, it is possible to manage the concentration and morphological properties of copper-containing nanoparticles so that they cause no harm to normal cells [11]. Additionally, it is rare for nanoparticles applied to dental materials to enter the body. Adding adhesives containing copper nanoparticles will not further cytotoxicate the pulp or oral soft tissues, according to multiple research studies. As an example, adhesives containing copper iodide nanoparticles coated with polyacrylic acid were created by Sabatini et al. [12]. Once the adhesive has been aged for one year, it will be able to effectively inhibit the growth of bacteria without compromising its bond strength or cytotoxicity.

Data on the bonding characteristics of CAD/CAM resin composites are lacking in the scientific literature because these materials are still in their infancy [13]. It is well known that the composition and structure of a material determine its bonding properties; this is especially true with conventional resin composites and CAD/CAM resin blocks [14]. Aladag et al. [15] assessed the influence of both thermal cycling and surface treatment techniques on the bonding efficacy of multistep adhesive resin cements for CAD/CAM resin blocks. They demonstrated that comparable bond strength values can be attained using either hydrofluoric acid or sandblasting with Al_2_O_3_. Moreover, Senol et al. [16] investigated the bond strength of conventional resin-based adhesive cement and self-adhesive resin cement for CAD-CAM resin blocks, revealing that the application of thermal cycling resulted in a reduction in the bond strength values of both types of resin cements for CAD-CAM resin blocks.

Currently, research on the application of copper for dentin bonding pretreatment is limited. However, one study conducted by Pan et al. [17] proved that dentin microtensile bond strength (μTBS) was enhanced by a copper-based preparation when used with universal adhesives. Thus, this study aims to investigate the impact of a CuSO_4_ + K_2_HPO_4_ pretreatment on dentin–resin interface μTBS when bonded with a universal adhesive to conventional and CAD/CAM resin composites. The null hypothesis is that there is no significant difference in dentin μTBS with and without CuSO_4_ + K_2_HPO_4_ pretreatment when bonded with a universal adhesive to conventional and CAD/CAM resin composites.

## 2. Materials and Methods

### 2.1. Sample Size Calculation

To test the null hypothesis that no differences exist among the different groups that were investigated, a power analysis was carried out. Based on the results of Pan et al.’s [17] investigation, the minimum required sample size (n) was found to be 60, which is equivalent to 10 samples per group, using an alpha (α) level of 0.05 (5%), a beta (β) level of 0.2 (20%), and an effect size (ϋ) of 0.517. The G*Power software, version 3.1.9.7 (Heinrich-Heine-Universität Düsseldorf, Düsseldorf, Germany) was used to calculate the sample size [13]. Statistical analysis utilized each tooth as the statistical unit; the mean μTBS derived from the 10 sticks of each tooth represented the bond strength of that tooth.

### 2.2. Study Design and Ethical Approval

The present in-vitro experimental study was approved by the institution’s Research Ethics Committee (approval No. SCBR-363/2024 and approval date 1 December 2024). It was conducted at the College of Dentistry, Prince Sattam bin Abdulaziz University. All procedures followed the standards set by CONSORT [18]. Because this was an in vitro experiment, no Clinical Trials Registry registration was necessary.

### 2.3. Randomization, Grouping, and Blinding

A total of eighty recently extracted human molars were placed in transparent acrylic blocks to expose the crowns entirely and given numbers. Based on the dentin pretreatment, adhesion strategy, and type of resin composite, teeth were randomly allocated into eight equal groups (n = 10) as shown in Table 1 and Figure 1. Using Excel 2010 (Microsoft, Redmond, WA, USA), the lab assistant created a six-block randomization sequence. To hide the randomization sequence, opaque sealed envelopes were used. The observer was totally blinded to the experimental procedures; however, the operator was informed about the randomization sequence right before they began. Reliability tests to assess the intra-examiner reproducibility between the two investigators were performed. 

### 2.4. Specimen Preparation

Eighty human third molars were obtained from Department of Oral and Maxillofacial Surgery at the College of Dentistry, Prince Sattam bin Abdulaziz University. Teeth were taken from unhealthy patients with periodontal disease who had their teeth removed after their written informed consent had been obtained. The teeth were preserved in a 0.5% chloramine solution; these were utilized in a period of one month. To expose the flat dentin surface, a diamond saw (IsoMet 4000, Buehler, Lake Bluff, IL, USA) was utilized to horizontally cut the occlusal enamel at a level 1 mm below the dentin–enamel junction. An Olympus light microscope (Olympus SZ61, Munster, Germany), used at a magnification of 20×, was used to verify the exact removal of the occlusal enamel.

### 2.5. Restorative Procedures

#### 2.5.1. Conventional Composite

The dentin surfaces of the tooth samples that were part of the etch-and-rinse groups were first polished and then acid-etched for 20 s using 37% phosphoric acid (Scotchbond Etchant, 3M Oral Care; St. Paul, MN, USA). The samples were then washed with deionized water for 20 s. No etching was done on the dentin surface for the self-etching groups. Then, either a new dentin-bonding pretreatment with 0.015 mol/L copper sulfate (CuSO_4_) and 0.01 mol/L dipotassium hydrogen phosphate (K_2_HPO_4_) was used (experimental group; Figure 2), or no treatment was carried out (control group). CuSO_4_ was mixed with distilled water at 37 °C to form a 0.015 mol/L blue CuSO_4_ solution (pH = 4). K_2_HPO_4_ was mixed with distilled water at 37 °C to form a 0.01 mol/L colorless liquid (pH = 7). In the control group, we used a tiny brush to gently remove excess water from the dentin surface while keeping it moist. The initial step was to apply 20 μL of CuSO_4_ solution to the dentin surface for 20 s in the experimental groups. Any surplus solution was then wiped away using a small, dry brush. The dentin surface was subsequently rubbed with a 20 μL K_2_HPO_4_ solution for 20 s. After rinsing with deionized water for 20 s, a microbrush was used to remove excess water and maintain moisture on the dentin surface.

Next, using either an etch-and-rinse or self-etching technique, the dentin surface was agitated for 20 s with the bonding adhesive Scotchbond Universal Plus (3M Oral Care, Seefeld, Germany). Using the incremental technique, a nano-filled resin composite was built. It was then light-cured for 20 s using a light-emitting diode curing unit (3M ESPE Elipar, Seefeld, Germany), which delivers 1200 mW/cm^2^ of light at 430–480 nm.

#### 2.5.2. CAD/CAM Resin Blocks

To ensure accurate surface scanning without interference from laser beam reflections, tooth specimens were first covered with an opaque scanning spray. Then, a digital scanner (Freedom, DOF Inc., Seoul, Republic of Korea) was used for the scanning process. For design reasons, the acquired image was loaded into the CAD program (Exocad DentalCAD Galway 3.0; Align Technology, San José, CA, USA), and a cement spacer size of 50 μm was established. Subsequently, occlusal restorations of a consistent 4 mm thickness were manufactured (Figure 3). 

Pressing the milling preview window’s button started the milling process. To start the wet milling operation, the 5-axis milling machine was turned on, and the Shufo hybrid ceramic disc (SHOFU Inc., Kyoto, Japan) was inserted into the spindle in a certain sequence. After the design was complete, the restoration STL file was sent into the milling machine (MC XL, Dentsply Sirona, Charlotte, NC, USA) in fine mode. After a series of cuts with 2.5 mm diamond burs, the disc was polished to eliminate any excess material at the connection point using a slow-speed diamond disk. Twenty specimens were milled in all (n = 20). To achieve the desired thickness, each restoration was polished using Kenda polishing tips (Kenda Unicus, Kenda AG, Vaduz, Liechtenstein), following the manufacturer’s recommendations. Multiple positions were used to measure the ultimate thickness with a caliper (lwanson’s caliper, Renfert, Hilzingen, Germany).

The interior surface of the specimen was etched for 60 s using a 5% hydrofluoric acid gel (VITA ADIVA CERAETCH, VITA Zahnfabrik, Bad Säckingen, Germany). After that, it was rinsed with water for 30 s, and then dried for 20 s using oil- and water-free air. It appeared as though the etched surface was frosty and dull. Phosphoric acid gel was used for cleaning, followed by rinsing and drying. A small amount of silane primer (RelyX^®^ Ceramic Primer, 3M/ESPE, St. Paul, MN, USA) was applied to the etched surfaces of the specimens using microbrushing. After 60 s, it was evenly spread with air to achieve a thin coat.

The next step was to apply and blast a single coat of Scotchbond Universal Plus universal adhesive on the fitting surface of the resin block specimens. Lastly, to make sure it cured evenly, it was light-cured for 20 s. The dentin surface was then agitated for 20 s with bonding adhesive Scotchbond Universal Plus using either an etch-and-rinse or self-etching approach. Following the manufacturer’s directions, the tips were used to apply dual-cured adhesive resin cement (RelyX Ultimate, 3-M ESPE, St. Paul, MN, USA) to the surfaces of milled hybrid ceramic restorations and tooth specimens. To facilitate cement flow from all directions, all restorations were placed softly on the specimens. Any extra cement was carefully removed after lightly curing the filling edges for 4–5 s on each side. The restorations were light-cured for 20 s on each side after they were established on the stumps. A water bath was used to keep the samples at 37 °C for 24 h after they were generated (Figure 4). Materials utilized in the current study are shown in Table 2.

### 2.6. Microtensile Bond Strength Test (μTBS)

The specimens were sectioned longitudinally through the bonded interface using an IsoMet diamond saw (IsoMetTM 4000, Buehler Ltd., Lake Bluff, IL, USA; Figure 5). Ten beams were obtained from each tooth. Each beam was secured to a testing jig using cyanoacrylate glue and then stressed until failure using a universal testing machine (Instron 4000, Canton, MA, USA) at a cross-head speed of 0.5 mm/mm. Microtensile bond strength was calculated in MPa.

### 2.7. Statistical Analysis

The results were statistically evaluated utilizing the Statistical Package for the Social Sciences (SPSS 25.0, IBM/SPSS Inc., Chicago, IL, USA). The Kolmogorov–Smirnov test at α = 0.05 was utilized to verify the normal distribution of the results. A three-way ANOVA test was used to estimate how the mean of a quantitative variable changed according to the levels of three independent variables, followed by Tukey’s post hoc test for multiple comparisons. *p*-values > 0.05 are statistically non-significant, *p*-values ≤ 0.05 are significant, and *p*-values ≤ 0.001 are highly significant.

## 3. Results

### 3.1. Microtensile Bond Strength Values

Intra-rater and inter-rater reliability for all outcomes measured were >0.80 kappa statistics. The Shapiro–Wilk test indicated that the data from all groups followed a normal distribution (*p* > 0.05). The μTBS values were shown to be not significantly affected by either the resin composite type (*p* > 0.05) or the adhesive strategy (*p* > 0.05) according to the three-way ANOVA results. Conversely, the μTBS was shown to be significantly affected by dentin pretreatment (*p* < 0.001; Table 3). Comparison between the study groups based on mean μTBS is shown in Table 4. Hence, significant differences were detected between the no dentin pretreatment (24.20 ± 4.54 MPa) and CuSO_4_ + K_2_HPO_4_ pretreatment (33.66 ± 5.22 MPa) using the etch-and-rinse adhesive strategy for nano-filled composites (*p* < 0.001). Additionally, significant differences were detected between the no dentin pretreatment (24.71 ± 4.33 MPa) and CuSO_4_ + K_2_HPO_4_ pretreatment (32.49 ± 4.92 MPa) using the etch-and-rinse adhesive strategy for CAD/CAM resin blocks (*p* < 0.001). Moreover, significant differences were detected between the no dentin pretreatment (21.20 ± 3.40 MPa) and CuSO_4_ + K_2_HPO_4_ pretreatment (30.31 ± 3.87 MPa) using the self-etching adhesive strategy for nano-filled composites (*p* < 0.001). Also, significant differences were detected between the no dentin pretreatment (23.89 ± 3.89 MPa) and CuSO_4_ + K_2_HPO_4_ pretreatment (31.22 ± 4.71 MPa) using the self-etching adhesive strategy for CAD/CAM resin blocks (*p* < 0.001; Figure 6 and Figure 7). For nano-filled composites, there was no discernible difference between the etch-and-rinse and self-etching groups (*p* > 0.001). For the CAD/CAM resin blocks, no significant difference was detected between the etch-and-rinse and self-etching groups (*p* > 0.001; Figure 7). In addition, a comparison of the nano-filled and CAD/CAM resin blocks without the dentin pretreatment (*p* > 0.001) and with the CuSO_4_ + K_2_HPO_4_ pretreatment (*p* > 0.001) found no statistically significant difference (Figure 8).

### 3.2. Failure Mode Analysis

The most prevalent type of fracture in all groups was a mixed failure, followed by an adhesive junction failure. The dentin and composite had minimal cohesive failures. Cohesive resin failures and adhesive junction failures occurred simultaneously throughout each mixed failure. Figure 9 and Figure 10 show representative stereomicroscope images of different failure modes. Figure 11 summarizes the distribution of the failure modes.

## 4. Discussion

Pretreatment of dentin with CuSO_4_ and K_2_HPO_4_ might render the entire hybrid layer more hydrophobic, which would improve dentin bonding. The results rejected the null hypothesis that there is no significant difference in dentin μTBS with and without CuSO_4_ + K_2_HPO_4_ pretreatment when bonded with a universal adhesive to conventional and CAD/CAM resin composites.

According to the literature, resin cements used in ceramic crowns should have a cement layer thickness of approximately 50–100 μm [19]. In addition, research has demonstrated that bonding characteristics are considerably diminished for a cement thickness ranging from 450 to 500 μm as a result of the residual tension caused by polymerization shrinkage [20]. For this reason, a cement spacer size of 50 μm was determined in this investigation.

In the present study, the μTBS significantly increased following CuSO_4_ and K_2_HPO_4_ pretreatment with nano-filled and CAD/CAM resin composites using both the etch-and-rinse and self-etching adhesive strategies. The bonding efficiency of different adhesives can be enhanced by the rapid formation of amorphous CaF_2_ nanoparticles in the demineralized dentin matrix following a simple sequential treatment of CaCl_2_ and NaF solutions. Endogenous, non-collagenous negatively charged groups are present in these nanoparticles [21].

Similarly, prior research demonstrated that nanoparticles produced by pretreatment of CaCl_2_ and K_2_HPO_4_ solutions enhanced dentin bonding characteristics. Like the aforementioned nanoparticles, Cu_3_(PO_4_)_2_ nanoparticles produced by Cu-P pretreatment may play a role in enhancing microtensile bond strength [21]. Theoretically, demineralized dentin might have its hydrophilicity mitigated by Cu-P pretreatment, and the resulting copper phosphate nanoparticles could provide structural support to the demineralized collagen fibrils. This structural support enhances the hybrid layer, and adhesive monomers are more effectively able to penetrate the interstices of the demineralized collagen fibrils network [8,22]. In the future, it will be necessary to thoroughly verify the exact process after Cu-P pretreatment using methods such as atomic force microscopy, solid surface potential, ion staining, water absorption property, and contact-angle tests.

Copper has characteristics that assist with dentin adhesion. Copper is essential for the activity of the collagen cross-linking enzyme lysyl oxidase [23]. The hybrid layer’s copper component strengthens the collagen network by acting as a cross-linking agent [22]. New evidence suggests that copper ions, which inhibit MMPs and can decrease MMP-2 synthesis, may regulate the activity of MMPs in oral tissues [24]. The impact of CuSO_4_ on collagen cross-linking and MMP inhibition will be subsequently validated. The findings of the present investigation align with those of Pan et al. [17], who demonstrated that copper-based pretreatment, when used with universal adhesives, enhanced the μTBS of dentin.

Cavity disinfectants based on CuSO_4_ have been widely employed in clinical practice due to their wettability and disinfecting effect [25]. This impact was probably caused by CuSO_4_’s ability to degrade the auto-associative propensity of water, which improves wettability in demineralized dentin, and dissociate strongly cross-linked collagen into a more distributed network of apparent fibrils [17]. Controversial findings persist despite claims that deproteinization causes dentin to become more porous and prone to abnormalities and anastomoses, which in turn increases bond strength [26].

The antibacterial properties of adhesives incorporating copper nanoparticles have been demonstrated in numerous investigations to be both long-lasting and effective. As an example, research conducted by Gutiérrez et al. [27] demonstrated that adhesive formulations can be enhanced with antibacterial capabilities when Cu nanoparticles are added at a concentration of 0.1 wt%. This enhancement was observed to occur without a compromise to the optical and mechanical properties of the adhesive solutions. Addition of 0.1 weight percent Cu nanoparticles to the adhesive can enhance the clinical efficacy of universal adhesive systems in non-carious cervical lesions without increasing cytotoxicity, according to a study by Matos et al. [28].

Dentin demineralization and smear layer removal are both aided by etching in the etch-and-rinse adhesive strategy, which increases the amount of monomer impregnation and exposes collagen fibrils; the result is a thick hybrid layer that is well integrated with the dentin surface [29,30]. This enhances the amount of monomer impregnation and reveals the collagen fibrils [31]. A downside is that it lowers the calcium phosphate concentration, which is practically nonexistent in hydroxyapatite crystals after etching [32]. Therefore, the primary diffusion-based bonding mechanism of the etch-and-rinse adhesive method for dentin is the hybridization or penetration of the resin into the exposed collagen fibril scaffold. Actual chemical bond formation is very unlikely due to the low affinity of the MDP monomers’ functional groups for the hydroxyapatite-depleted collagen [33,34].

Conversely, in the self-etching adhesive strategy, dentin priming and conditioning are accomplished by acidic resin monomers dissolving the smear layer, but the calcium phosphate in the hydroxyapatite surface remains intact [35,36]. The improved chemical link between the adhesive and the dentin substrate is a result of this closer chemical interaction with the functional groups of the MDP monomers [37]. Hence, the μTBS results of the Scotch Bond Universal Plus adhesive investigation showed that both etching adhesive strategies produced similar outcomes, a conclusion that is in line with previous research [38,39,40].

The self-etching and etch-and-rinse groups did not differ significantly from one another in this study. One component of Scotch Bond Universal Plus adhesive is a monomer of phosphate acid, which forms “nano-layering” on hydroxyapatite by chemically bonding to it. This results in the formation of hydrolytically stable calcium salts [41]. The similar adhesion–decalcification process also occurs when using Scotch Bond Universal Plus adhesive on apatite surfaces due to the presence of a polyalkenoic acid copolymer in this product [41]. Therefore, chemical adhesion and micromechanical retention through resin monomer diffusion were both enhanced by the bonding mechanisms of Scotch Bond Universal Plus adhesive.

In this investigation, we found no statistically significant difference between the self-etching and etch-and-rinse groups. Scotch Bond Universal Plus adhesive contains a phosphate acid monomer that chemically bonds to hydroxyapatite, forming hydrolytically stable calcium salts in the form of “nano-layering” on hydroxyapatite [41]. Additionally, Scotch Bond Universal Plus adhesive contains a polyalkenoic acid copolymer in its composition that interacts with apatite substrates following the same adhesion–decalcification reaction [42]. Consequently, the Scotch Bond Universal Plus adhesive facilitated micromechanical retention through the diffusion of resin monomers and chemical adhesion [43,44].

Munoz et al. [45] states that the bonding strength of MDP to the dentin substrate is negatively affected by the polyalkenoic acid copolymer because it hinders the chemical bond. In hydroxyapatite crystals, it competes with the MDP functional monomer for Ca-bonding sites and might even stop the monomers from approximation during polymerization. In addition, research has shown that 2-hydroxyethyl methacrylate can bind to the calcium in hydroxyapatite, which weakens the bond strength of MDP to dentin [46,47,48].

The bond strength findings were validated by the fracture beam analysis, which demonstrated an increased incidence of mixed failure in all groups and a reduced occurrence of cohesive failure in both the composite and dentin. Substrate fracture level is a good indicator of adhesive retentive strength, according to previous studies [17,49,50]. This study’s bond strength estimates were thus corroborated by the mode of failure. This agrees with the findings of Molla et al. [51], who also discovered that, at substantial bond strengths, most bond failure modes are admixed.

Furthermore, a comparison of nano-filled and CAD/CAM resin blocks, both without dentin pretreatment and with CuSO_4_ + K_2_HPO_4_ pretreatment, revealed no statistically significant difference. Hussain et al. [52] performed a study to compare conventional resin blocks with CAD/CAM resin blocks. They determined that the CAD/Cam resin blocks surpassed traditional composites in mechanical properties and monomer elution, although they exhibited concerning outcomes related to cytotoxicity. Mechanical research studies indicate that CAD/CAM resin blocks have superior characteristics compared to conventional composites and are equivalent to enamel and dentine. The filler content, morphology of filler particles, and filler composition primarily align with the producer’s specifications and resemble conventional composite materials [52].

Nevertheless, additional research into the bonding’s endurance and potential processes is required. Findings from this research will pave the way for novel approaches to enhancing the dentin–adhesive interface. Additionally, future research may adjust the aforementioned components to include aspects such as scanning electron microscopy, energy-dispersive X-ray, Fourier transform infrared spectroscopy, and atomic force microscopy. A more thorough investigation of their long-term degradation, particularly under in vivo conditions, is required. The main drawback of the present study is that in vitro investigations cannot replicate all the specific conditions to which a restoration is subjected in the oral cavity. To properly understand the differences between dentin pretreatment agents over time, it is crucial to evaluate the bonding performance of aged specimens using thermocycling or long-term water storage. This is given that our study only looked at immediate bond strength, but prior research has shown that there are negative effects associated with this thermocycling and long-term water storage.

## 5. Conclusions

Within the limitations of the study, dentin’s immediate μTBS was enhanced by a pretreatment using CuSO_4_ and K_2_HPO_4_ solutions when used with nano-filled and CAD/CAM resin blocks. Furthermore, the adhesion of Scotch Bond Universal Plus adhesive to dentin remains largely consistent regardless of whether the etch-and-rinse or self-etching adhesive technique is adopted.

## 6. Clinical Relevance

Dentin pretreatment with CuSO_4_ and K_2_HPO_4_ proved to be effective, which could enable general practitioners to use them in dental practice routinely.

## Figures and Tables

**Figure 1 jfb-16-00197-f001:**
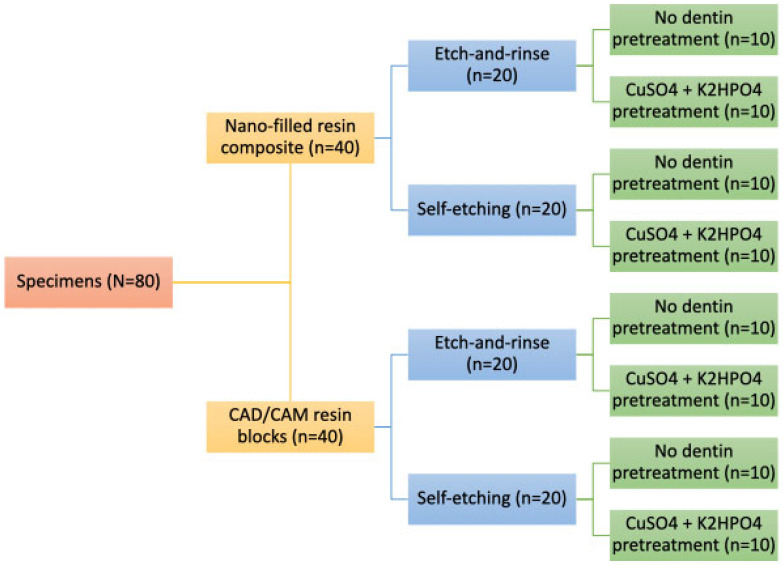
Schematic flow diagram showing the study design.

**Figure 2 jfb-16-00197-f002:**
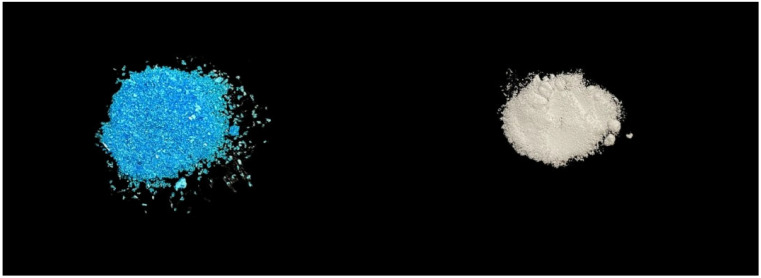
Images showing copper sulfate (CuSO_4_; blue) and dipotassium hydrogen phosphate (K_2_HPO_4_; white) powders.

**Figure 3 jfb-16-00197-f003:**
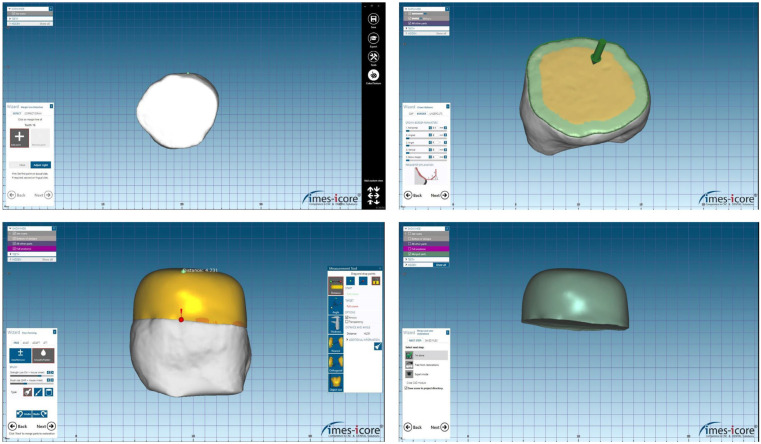
Images showing the design of specimens with 4 mm thickness using CAD software (Exocad DentalCAD Galway 3.0; Align Technology, San José, CA, USA).

**Figure 4 jfb-16-00197-f004:**
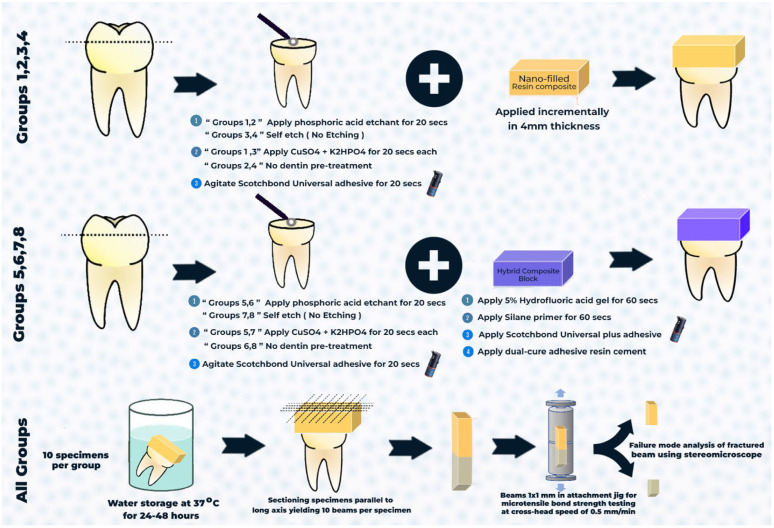
Schematic diagram shows step-by-step procedures, including surface treatment, composite application, storage, specimen sectioning, beam fixation in the attachment jig, and failure mode analysis. Group 1, no dentin pretreatment + etch-and-rinse + nano-filled composite; Group 2, CuSO_4_ + K_2_HPO_4_ pretreatment + etch-and-rinse + nano-filled composite; Group 3, no dentin pretreatment + self-etching + nano-filled composite; Group 4, CuSO_4_ + K_2_HPO_4_ pretreatment + self-etching + nano-filled composite; Group 5, no dentin pretreatment + etch-and-rinse + CAD/CAM resin blocks; Group 6, CuSO_4_ + K_2_HPO_4_ pretreatment + etch-and-rinse + CAD/CAM resin blocks; Group 7, no dentin pretreatment + self-etching + CAD/CAM resin blocks, and Group 8, CuSO_4_ + K_2_HPO_4_ pretreatment + self-etching + CAD/CAM resin blocks.

**Figure 5 jfb-16-00197-f005:**
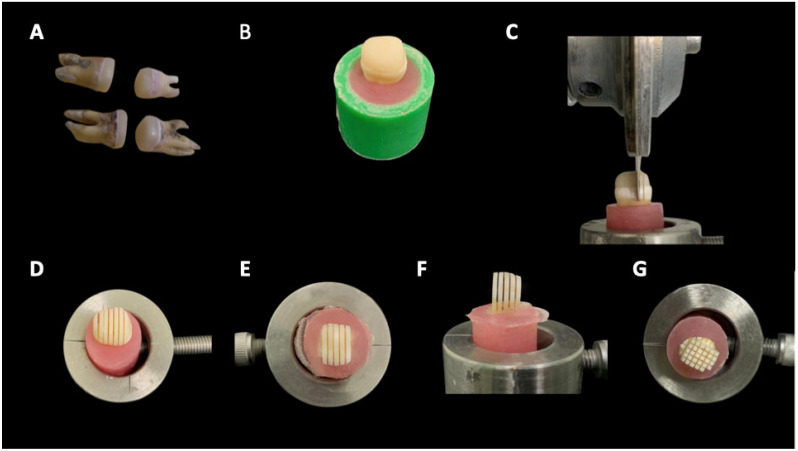
Representative images showing specimen-cutting procedures for obtaining beams for μTBS testing: (**A**), specimens (10 specimens per group); (**B**), specimen fixed in acrylic mold; (**C**), lateral view showing IsoMet saw-sectioning in one direction for CAD/CAM resin; (**D**), occlusal view for sectioning in one direction for CAD/CAM resin; (**E**), occlusal view for sectioning in one direction for conventional resin; (**F**), lateral view for sectioning in one direction for conventional resin; and (**G**), occlusal view for sectioning in opposite direction for conventional resin to obtain beams 1 × 1 mm thickness.

**Figure 6 jfb-16-00197-f006:**
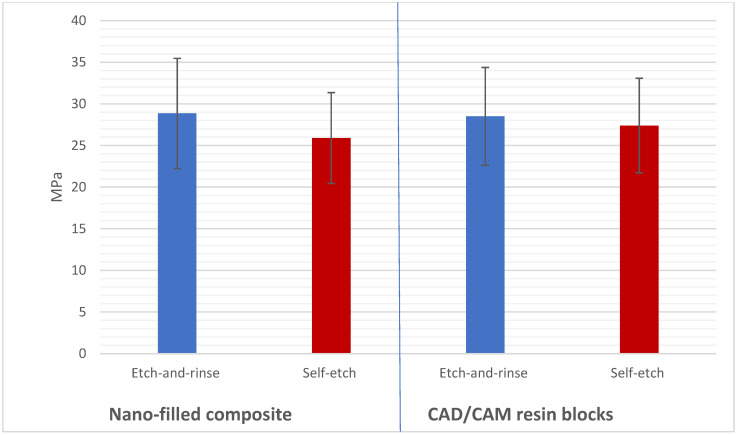
Graphically drawn mean values ± standard deviations showing the effect of resin composite and adhesive strategy on μTBS in MPa.

**Figure 7 jfb-16-00197-f007:**
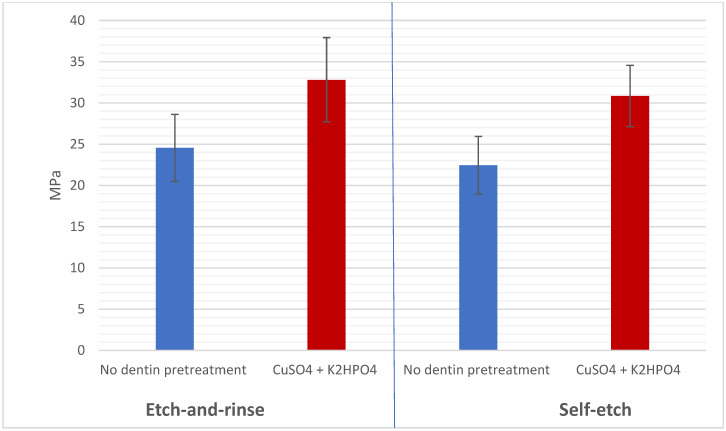
Graphically drawn mean values ± standard deviations comparing the effect of adhesive strategy and dentin pretreatment on μTBS in MPa.

**Figure 8 jfb-16-00197-f008:**
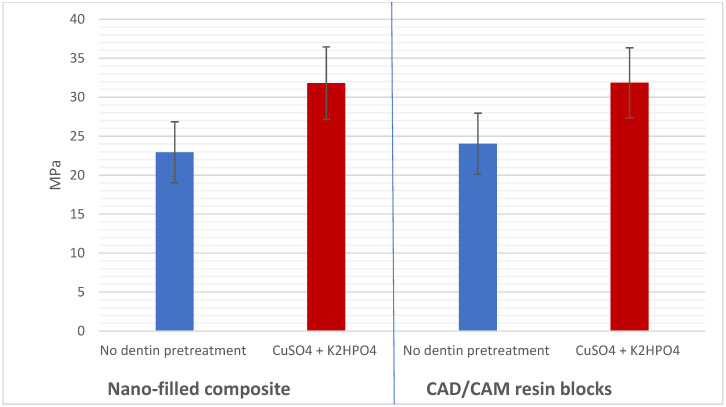
Graphically drawn mean values ± standard deviations showing the effect of resin composite and dentin pretreatment on μTBS in MPa.

**Figure 9 jfb-16-00197-f009:**
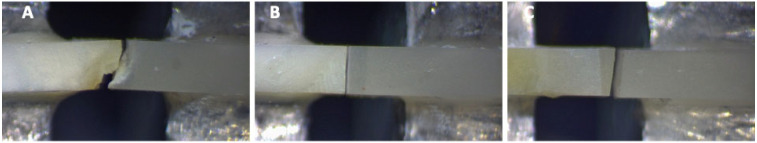
Representative stereomicroscopic images depicting various failure modes: (**A**) mixed failure; (**B**) adhesive junction failure; (**C**) cohesive failure in composites.

**Figure 10 jfb-16-00197-f010:**
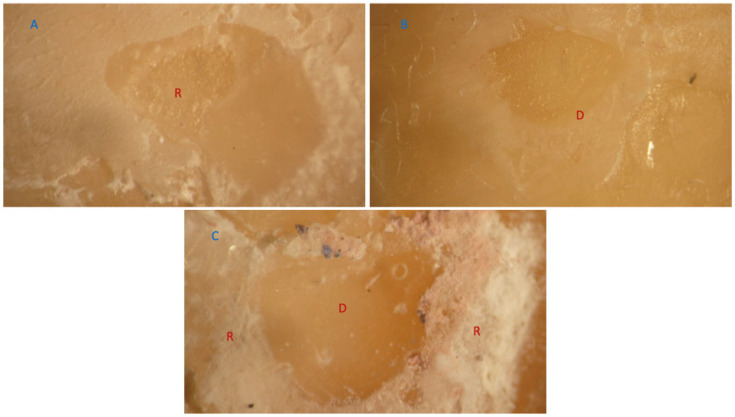
Stereomicroscope images of different failure modes: (**A**) cohesive failure in composite; (**B**) adhesive junction failure; (**C**) mixed failure; D, dentin; R, resin.

**Figure 11 jfb-16-00197-f011:**
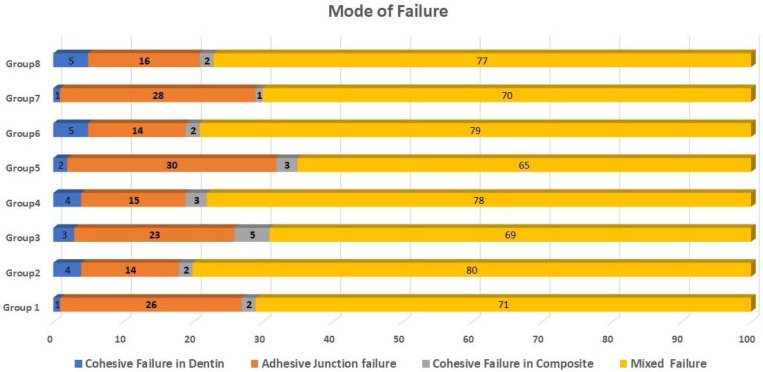
Distribution of failure modes.

**Table 1 jfb-16-00197-t001:** Dentin pretreatment and adhesive strategy performed in nano-filled and CAD/CAM resin composite groups.

Groups	Dentin Pretreatment	Adhesive Strategy	Resin Composite
Group 1	No dentin pretreatment	Etch-and-rinse	Nano-filled composite
Group 2	CuSO_4_ + K_2_HPO_4_	Etch-and-rinse	Nano-filled composite
Group 3	No dentin pretreatment	Self-etching	Nano-filled composite
Group 4	CuSO_4_ + K_2_HPO_4_	Self-etching	Nano-filled composite
Group 5	No dentin pretreatment	Etch-and-rinse	CAD/CAM resin blocks
Group 6	CuSO_4_ + K_2_HPO_4_	Etch-and-rinse	CAD/CAM resin blocks
Group 7	No dentin pretreatment	Self-etching	CAD/CAM resin blocks
Group 8	CuSO_4_ + K_2_HPO_4_	Self-etching	CAD/CAM resin blocks

**Table 2 jfb-16-00197-t002:** Composition, commercial name, and manufacturer of materials utilized in the current study.

Material	Commercial Name and Manufacturer	Composition
CAD/CAM resin block	SHOFU Inc., Kyoto, Japan	Triethylene glycol dimethacrylate, urethane dimethacrylate, silica powder, zirconium silicate, fine particulate silica and colorant
Nano-filled resin compsite	Filtek Z350 XT, 3M Oral Care, St. Paul, MN, USA	Urethane dimethacrylate, Bisphenol-A-glycidyl methacrylate, ethoxylated bisphenol-A dimethacrylate, in addition to trace amounts of triethylene glycol dimethacrylate and nanoagglomerated nanoparticles of zirconium/silica particles with sizes ranging from 0.6 to 1.4 mm, accounting for 78.5% of the total weight (59.5% of the volume)
Adhesive resin cement	RelyX Ultimate, 3-M ESPE, St. Paul, MN, USA	Silane-treated glass powder, 2-methyl, 1,10-[1-(hydroxymethyl) 1,2-ethanediyl] ester, 2-propenoic acid, TEGDEMA, silane-treated silica, oxide glass chemicals, sodium persulfate, tert-butyl peroxy-3,5,5 trimethylhexanoate, reaction products with 2-hydroxy-1,3-propanediyl dimethacrylate and phosphorus oxide, copper (II) acetate monohydrate, silane-treated silica, substituted dimethacrylate, 1,12-dodecane dimethacrylate, -benzyl-5-phenyl-barbic-acid, sodium p-toluenesulfinate, 2-propenoic acid, calcium salt, calcium hydroxide, and titanium dioxide
Scotchbond Universal Plus Adhesive	Surefil^®^ SDR™flow, Dentsply Caulk, Milford, DE, USA	2-hydroxyethyl methacrylate, 2-propenoic acid, 10-methacryloyloxydecyl dihydrogen phosphate, 1,3-benzenediol 2-(2-hydroxyethoxy) ethyl 3-hydroxypropyl diethers, 2-methyl-, 3 triethoxysilyl) propyl ester, reaction products with silica and 3-aminopropyl triethoxysilane, ethanol, water, camphorquinone, a copolymer of acrylic and itaconic acid, and copper (II) acetate monohydrate

**Table 3 jfb-16-00197-t003:** Three-way ANOVA for μTBS in MPa.

Source	Sum of Squares	df	Mean Squares	F	*p*-Value	Partial Eta Squared (Effect Size)
Corrected Model	1497.787 ^a^	7	213.970	12.073	0.000	0.540
Intercept	61,217.113	1	61,217.113	3453.986	0.000	0.980
Dentin pretreatment	1386.113	1	1386.113	78.207	0.000	0.521
Adhesive strategy	82.012	1	82.012	4.627	0.035	0.060
Resin composite	6.613	1	6.613	0.373	0.543	0.005
Dentin pretreatment * adhesive strategy	0.113	1	0.113	0.006	0.937	0.000
Dentin pretreatment * resin composite	5.513	1	5.513	0.311	0.579	0.004
Adhesive strategy * resin composite	17.113	1	17.113	0.966	0.329	0.013
Dentin pretreatment * Adhesive strategy * resin composite	0.313	1	0.313	0.018	0.895	0.000
Error	1276.100	72	17.724			
Total	63,991.000	80				
Corrected Total	2773.887	79				

^a^. R squared = 0.540; adjusted R squared = 0.495. *: Statistically significant (*p* ≤ 0.05).

**Table 4 jfb-16-00197-t004:** Microtensile bond strength (MPa) data (mean ± SD) in all groups.

Dentin Pretreatment + Adhesive Strategy	Resin Composite		
	Nano-Filled Resin Composite Mean ± SD	CAD/CAM Resin Block Mean ± SD	*p*-Value
No dentin pretreatment			
Etch-and-rinse	24.20 ± 4.54 ^Ac^	24.71 ± 4.33 ^Ac^	1.000
Self-etching	21.20 ± 3.40 ^Ac^	23.89 ± 3.89 ^Ac^	1.000
CuSO_4_ + K_2_HPO_4_ Dentin pretreatment			
Etch-and-rinse	33.66 ± 5.22 ^Aa^	32.49 ± 4.92 ^Aa^	1.000
Self-etching	30.31 ± 3.87 ^Ab^	31.22 ± 4.71 ^Aab^	1.000

Distinct upper-case letters in the same row signify a significant difference between groups; distinct lower-case letters in the same column denote a significant difference between groups (Tukey’s test, *p* < 0.05).

## Data Availability

The original contributions presented in the study are included in the article, further inquiries can be directed to the corresponding author.

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
