# Peer review of "Impact of a Novel Pretreatment on Bond Strength of Universal Adhesive to Conventional and CAD/CAM Resin Composites: In Vitro Study"

_jfb, 2025, doi:10.3390/jfb16060197_

Round 1
Reviewer 1 Report
Comments and Suggestions for Authors
The study addresses an important and timely topic in adhesive dentistry by evaluating a copper-based dentin pretreatment. The manuscript is generally well-organized and clearly written, with a solid experimental design. However, I have identified a couple areas that would benefit from clarification or further elaboration to strengthen the rigor and presentation of the study:
- Clarity of hypothesis and objectives
- The stated hypothesis in the abstract and introduction is vague and should be clearly framed as a null hypothesis. For example: “There is no significant difference in μTBS between dentin with and without CuSOâ‚„ + Kâ‚‚HPOâ‚„ pretreatment when bonded with a universal adhesive.”
- Mechanistic claims require supporting data
- While the discussion proposes that CuSOâ‚„ + Kâ‚‚HPOâ‚„ enhances bonding through nanoparticle formation and improved hydrophobicity, no direct surface characterization (SEM, EDX, FTIR, contact angle measurements) was performed.
- Please consider qualifying these mechanisms as hypotheses or future directions.
- Suggest including surface characterization in future studies to validate these assumptions.
- Surface analysis is missing
- Adding imaging or spectroscopy (e.g., SEM, AFM, or XPS) could greatly strengthen your claims about dentin modification and hybrid layer integrity. This is particularly relevant for a study proposing a novel pretreatment mechanism.
- Figures and tables
- Ensure figure legends clearly define sample sizes and abbreviations.
- Table 2 contains minor typographical errors: “Commerial” → “Commercial”; “voliume” → “volume”.
- Consider using a schematic flow diagram summarizing the 8 experimental groups to facilitate reader understanding.
- Discussion enhancements
- The section comparing self-etch vs. etch-and-rinse performance is insightful, particularly regarding MDP and Vitrebond copolymer interactions. However, be cautious not to overstate conclusions in the absence of direct compositional analysis of the bonding interface.
- Consider briefly discussing the clinical relevance and translational potential of the Cu-P pretreatment in long-term scenarios.
- Language and style
- Several sections would benefit from minor editing for grammar and clarity, such as:
- “The μTBS was significantly increased when CuSOâ‚„ and Kâ‚‚HPOâ‚„ pretreatment were utilized…” → consider simplifying to: “μTBS significantly increased following CuSOâ‚„ and Kâ‚‚HPOâ‚„ pretreatment…”
- Remove redundant phrases like "the purpose of this research was to develop an entirely novel pretreatment..."
- Limitations
- Please explicitly state the study’s limitations in the discussion or conclusion (e.g., in vitro design, lack of long-term degradation testing).
Author Response
Dear respected Assigned Editor of JFB,
Revision of manuscript jfb-3638485 titled: ‘‘Impact of a novel pretreatment on micro-tensile bond strength of an innovative universal adhesive to conventional and CAD/CAM resin composites’’
The authors of this manuscript highly appreciate all the reviewers’ constructive comments and accordingly, we have extensively revised the manuscript text, and we are ready for any further comments. Attached is the summary of our response to the reviewer/s opinion.
General response:
We respect all the reviewers’ opinions and appreciate all the reviewers’ constructive comments. Accordingly, we did our best to respond positively to all comments and make all possible changes. We will respond to any further changes that may be needed. (Changes in the revised manuscript are highlighted in BLUE color)

Reviewer 2 Report
Comments and Suggestions for Authors
Thank you for the opportunity to review your manuscript entitled “Impact of a Novel Pretreatment on Micro-Tensile Bond Strength of an Innovative Universal Adhesive to Conventional and CAD/CAM Resin Composites.” The topic is timely and clinically relevant; with revision, the work could make a valuable contribution to the literature. My detailed observations follow, grouped by manuscript section so you can address each point systematically.
First, I encourage you to shorten and sharpen the title and abstract. A concise title that highlights the pretreatment, adhesive, and in-vitro nature of the study will improve discoverability. In the abstract, please report the absolute μTBS means ± SD (or the % increase) so readers can immediately gauge the practical magnitude of your findings.
In the Introduction, the clinical challenge of bonding to CAD/CAM blocks is well framed, yet the background relies heavily on manufacturer claims. Please integrate more peer-reviewed data—particularly 2024–2025 accelerated-aging studies—so the scientific context rests on independent evidence. Clarifying the null hypothesis in a single, positive statement (e.g., “the pretreatment will not alter μTBS…”) will also improve readability.
For the Materials and Methods, the work is best described as an in-vitro experimental study rather than a “randomized controlled trial.” Please specify whether statistical analysis used the mean of all sticks from each tooth (recommended) or treated every stick as an independent specimen (risking pseudoreplication). Describing the pH, preparation, and stability of the CuSOâ‚„/Kâ‚‚HPOâ‚„ solution, together with the exact adhesive protocols, will enhance reproducibility. Finally, consider adding thermocycling (e.g., 10 000 cycles, 5 °C/55 °C) or long-term water storage to strengthen the clinical relevance of your results.
With respect to statistical analysis, a three-way ANOVA is appropriate, but please report the post-hoc comparisons with a multiple-comparison correction (Tukey-HSD or Bonferroni). Including effect sizes (η² or Cohen’s d) will allow readers to gauge practical significance. Note that a negative “F” value cannot occur; please correct the table headers or statistics accordingly.
In the Results and Discussion, the finding that the pretreatment increased μTBS by approximately 8 MPa in both composite types is promising; however, the absence of differences between the CAD/CAM and nano-composite groups requires deeper discussion. Relate these observations to filler content, elastic modulus, and previously reported bonding mechanisms. Please remove subjective adjectives such as “revolutionary” or “entirely novel,” and highlight instead the mechanistic insight your data provide. Because your durability assessment is limited to 24 h storage, be sure to emphasize this limitation and outline plans for future long-term testing.
Regarding figures and tables, ensure that all group names are written in full and that color choices remain legible for readers with color-vision deficiency. Add scale bars to micrographs and confirm that row or column percentages sum to 100 % where appropriate.
Finally, in the reference list, replace manufacturer leaflets with peer-reviewed sources wherever possible, and add at least two recent (2024–2025) papers on universal-adhesive durability to demonstrate awareness of the latest work in the field.
I hope these comments prove constructive and assist you in strengthening the study. I look forward to reviewing a revised version and remain confident that, once the methodological and reporting issues are addressed, the manuscript will meet the standards of Journal of Functional Biomaterials.
Comments on the Quality of English LanguageThe manuscript is generally understandable; however, the quality of English can be improved to enhance clarity and scientific tone. Several sentences are overly wordy or include subjective expressions such as “revolutionary” or “entirely novel,” which are not appropriate in scientific writing. Additionally, some methodological descriptions (e.g., pretreatment protocol, statistical analysis) would benefit from clearer, more concise phrasing. A moderate professional language edit is recommended.
Author Response

(The authors gave the same response as above.)

Reviewer 3 Report
Comments and Suggestions for Authors
This study aimed to investigate the impact of a novel pretreatment
on the mTBS of an innovative universal adhesive to conventional and CAD/CAM resin
composites. I appreciate the subject and consider it useful for the dental community.
However, some aspects should be improved:
The title does not clearly define the core content. Innovative universal adhesive is too vague.
Abstract:
Key details about the materials, methods, and quantitative results are missing. For example, no release kinetics values or mechanical properties are properly mentioned.
Introduction:
Why not use an experimental adhesive?
What about copper toxicity?
Reference 4 is wrongly cited.
Very strange sentence “There is a dearth of data on the bonding characteristics of CAD/CAM resin composites in the scientific literature because these materials are still in their infancy “. Did you use AI to write your text? Please double-check the language.
You mention the aim of evaluating“ novel pretreatment”. There is nothing about it in the introduction, only about the adhesive composition.
Methods:
According to Pan et al.,[12], a minimum of 60 samples is required per group or in total?
Polymer synthesis and metal loading protocols are underdescribed.
Why were these specific tests chosen? What standards were followed?
There is no mention of replicates or significance thresholds.
Are you testing the bonding between dentin/cement or between cement/restoration?
Results:
Some figures are poorly labelled.
Your text mostly describes figures without extracting what is clinically relevant.
Figure 8 should be replaced with an internal view of the failure. Observing only the lateral portion of the beams does not provide enough information to determine the failure mode
Discussion:
Please add the possible biological effect expected when using your new approach.
Aging was not simulated. Please discuss it.
What about the cement layer thickness? How was it controlled? Was is the effect on it?PMID: 34576376
Conclusion:
Only the immediate bond strength was analysed. Make it clear.
Comments on the Quality of English Language
Some sentences are clearly distorted or written with unusual synonyms. Please check for AI-generated text.
Author Response

(The authors gave the same response as above.)

Round 2
Reviewer 2 Report
Comments and Suggestions for Authors
Thank you for your thorough and constructive responses to all reviewer comments. The revised manuscript is significantly improved in structure, clarity, and methodological transparency. The additions to the statistical section, clarification of the experimental design, and the removal of subjective language have enhanced the overall quality of the study. I believe the current version is acceptable for publication.
Comments on the Quality of English LanguageThe language has been improved and is now clear and appropriate for scientific publication. Minor polishing could still be beneficial, but the overall clarity and readability are satisfactory.
Author Response
We appreciate the reviewer’s constructive comment. Therefore, the whole manuscript was English-proofread to correct typographical errors, grammar, and to ensure clarity of the entire manuscript.
Reviewer 3 Report
Comments and Suggestions for Authors
Thanks for the new version.
Author Response
Great thanks for your valuable contribution